# Model for Risk Calculation and Reliability Comparison of Level Crossings

**DOI:** 10.3390/e23091230

**Published:** 2021-09-19

**Authors:** Pamela Ercegovac, Gordan Stojić, Miloš Kopić, Željko Stević, Feta Sinani, Ilija Tanackov

**Affiliations:** 1Faculty of Technical Sciences, University of Novi Sad, Trg Dositeja Obradovića 6, 21000 Novi Sad, Serbia; pamela.ercegovac@uns.ac.rs (P.E.); gordan@uns.ac.rs (G.S.); ilijat@uns.ac.rs (I.T.); 2Faculty of Transport and Traffic Engineering Doboj, University of East Sarajevo, Vojvode Mišića 52, 74000 Doboj, Bosnia and Herzegovina; zeljkostevic88@yahoo.com or; 3Faculty of Applied Sciences, State University of Tetova, Ilindenska, 1000 Tetovo, North Macedonia; feta.sinani@unite.edu.mk

**Keywords:** risk, reliability, level crossings, queueing theory, accident, entropy

## Abstract

There is not a single country in the world that is so rich that it can remove all level crossings or provide their denivelation in order to absolutely avoid the possibility of accidents at the intersections of railways and road traffic. In the Republic of Serbia alone, the largest number of accidents occur at passive crossings, which make up three-quarters of the total number of crossings. Therefore, it is necessary to constantly find solutions to the problem of priorities when choosing level crossings where it is necessary to raise the level of security, primarily by analyzing the risk and reliability at all level crossings. This paper presents a model that enables this. The calculation of the maximal risk of a level crossing is achieved under the conditions of generating the maximum entropy in the virtual operating mode. The basis of the model is a heterogeneous queuing system. Maximum entropy is based on the mandatory application of an exponential distribution. The system is Markovian and is solved by a standard analytical concept. The basic input parameters for the calculation of the maximal risk are the geometric characteristics of the level crossing and the intensities and structure of the flows of road and railway vehicles. The real risk is based on statistical records of accidents and flow intensities. The exact reliability of the level crossing is calculated from the ratio of real and maximal risk, which enables their further comparison in order to raise the level of safety, and that is the basic idea of this paper.

## 1. Introduction

There is a large and significant number of elements that define and influence the concept of safety. Therefore, it is difficult to express these terms in any engineering quantity or certain indicator that would be completely unambiguous and clear. Therefore, the usual approach to safety analysis is deductive, starting from the phenomenon of an accident in which its numerous attributes are synthesized.

Accidents at road crossings occur in disproportionate ratios of kinetic energies, with a pronounced dominance of the consistency of railway vehicles. Large undesirable balances of accidents at road crossings have direct consequences: Loss of human lives, physical disabilities and psychological trauma that are prevalent in road traffic at the expense of destroyed vehicles and signaling and safety devices. Indirect costs include disruptions in the timetable of railway traffic and suspension of road traffic. Therefore, the topic of safety at road crossings is prominent.

This paper proposes a new mathematical model that, by calculating the maximal risk, introduces a new reference value in the calculation of the reliability of level crossings. The given model uses the conventional notation of the queuing theory, Kendall notation [1] or Kendal-Lee notation [2,3]. The model is based on a theoretical consideration of the chaotic movement of road and railway vehicles in the queuing system of the level crossing. The concept of chaos is designed by eliminating common priorities and a complete information deficit, without the use of signalization. The maximal stochastic entropy in conditions of chaos is generated by the exponential distribution of access to and service of road and railway vehicles. The described conditions generate a number of accidents at the virtual level crossing that corresponds to the maximal risk. Regardless of the intensities of access and service, the value of maximal risk is convergent!

In the defined engineering framework, from the deterministic to the stochastic limit of the abstract number of accidents, there is a proportion for estimating and comparing the risk of accidents at level crossings. The calculation of this proportion is based on real, statistically recorded accidents, which enables the determination of safety levels for each level crossing individually. A new approach in the assessment and evaluation of traffic safety at road crossings is the basic contribution to safety research at the intersections of railways and road traffic.

## 2. Models for Predicting Accidents and Incidents at Level Crossings—A Review of the Literature

Safety at level crossings is one of the most critical issues to be addressed on railways [4,5,6]. In 28 countries of the European Union in 2012, there were more than 118,000 level crossings, which corresponds to an average of five crossings per 10 km of railway [7]. Accidents at road crossings in Europe result in more than 300 deaths each year [5], and in some European countries, they account for about 50% of all rail accidents [8,9]. In France, across the 30,000 km of the railway network, there are 18,000 level crossings, which are crossed daily, by an average of about 16 million vehicles [10]. Regardless of the measures taken to improve safety at road crossings in France, 100 accidents were recorded in which 25 people lost their lives in 2014, and further reductions are being sought [11]. In 2012, the US Federal Railroad administration conducted an investigation in North Carolina and considered only the probability of fatal accidents at 44 crossings where security levels were raised [12]. For the period before and after the applied measures, it was estimated that 1.5 lives were saved, as a result of the improvement of the level of security of level crossings. This method estimated that the improvements implemented by 2008 would reduce the number of deaths by about 0.39 per year. Most traffic accident models have been developed based on statistical regression techniques. As one of the most basic methods, the Poisson regression model has been widely used for collision analysis and accident modeling [13,14]. Poisson’s distribution model is usually used to predict the number of accidents, due to the non-negative, discrete and random characteristics of accidents. In most cases, accident data are too dispersed, and although they have a significant advantage in precision modeling [15], Poisson models are inadequate for use with over-dispersed data, which have a variance greater than the mean [16]. Other models are in use as well, including Poisson-lognormal models [17,18,19]. Several authors have also suggested the use of a negative binomial model (NB) or a Poisson-gamma model [15,20,21,22,23,24,25,26,27] because it is much more suitable for datasets with “greater dispersion, i.e., variance”. Such a model allows the inequality of mean and variance [28]. In addition, the Conway–Maxwell–Poisson distribution was introduced to model either excessively or insufficiently dispersed data [29,30,31]. Another source of excessive dispersion, in addition to hidden inhomogeneity, may be the high frequency of zeros in the sample, and this problem was solved by the Poisson model with excess zeros (ZIP) [32], and in further research, a negative binomial model with excess zeros proved to be a better choice (ZINB). Road components such as intersections or road segments, and in our case level crossings, have two conditions, and they are, perfect, in cases of almost perfect safety, and unsafe conditions [33]. The set of accident data may occasionally be insufficiently dispersed, i.e., the variance is less than the mean [34], where a negative binomial model has been recommended by many researchers [35]. Accident data collected for traffic safety studies often have unusual features: Low mean values of the samples, and due to the high cost of data collection, small sample sizes. When applying accident prediction models developed in other countries and areas of jurisdiction and under different conditions, there is no certainty that the model will be valid for locally analyzed conditions. A total of 22 different approaches in analyses and methods for determining the risk of accidents at road crossings have been identified in 12 countries around the world. The differences are reflected in their complexity and approach, which can be classified into four main groups based on the different complexity of computational algorithms [36,37]. The first group consists of the use of simple parameters that serve as tips for choosing the level of security in the following countries: India, Japan, Russia, Spain and Sweden. The second group consists of simple weighted factors that indicate the relative contribution of each parameter to the overall risk and are Australia (ALCAM model), Northern Ireland and New Zealand. The third group consists of complex weighted factors used in the following countries: Great Britain, Ireland and Spain (Failure Mode and Effects Analysis). The fourth group consists of statistical models based on statistical techniques of weight assessment (empirical terms) for parameters and are applied in Great Britain, Australia (Risk Assessment of Accident and Incident at Level crossings), Canada (GradeX), New Zealand and the USA (formulas for predicting the severity of an accident, GradeDEC.net). In some countries, risk modeling is based on the application of more advanced techniques, such as empirical Bayesian methods, FTA (Fault-Tree Analysis), ETA (Event-Tree Analysis), human factor analysis, etc. A large number of existing models are based on analyses of the occurrence of accidents that have occurred in the past, as well as monitoring the effects before and after the implementation of certain measures. The current version of the US DOT formula for accident prediction is presented in the Handbook for level crossing, revised second edition in 2007 by the Federal Highway Administration - FHWA) [38]. It was developed using data on accidents at national level crossings from 1981 to 1986. The formula has shortcomings [39] in weighting the contribution of various safety factors in estimating accident rates as well as inaccuracies in update formulas. Because it is based on data from all over the United States, it cannot take into account regional differences. The variables included in the models, or their coefficients, have not been modified since the publication of the original 1980 study, except for the third element of the model consisting of a normalizing constant factorized in the model just before the final collision prediction [40], and improvements should be made in identifying new variables.

A French group of authors developed a preliminary model for predicting accidents at level crossings, followed by an improved model based on the preliminary model [41]. It has been proven that the occurrence of accidents at level crossings is positively influenced by average daily railway traffic and speed limits on railways, average daily road traffic and the annual number of traffic accidents, road alignment, crossing width, crossing length and regional factors for observed level crossings. Although this model has only been applied to automatic railway crossings with two half-barriers and flashing lights, it can also be used at other crossings. In such a manner, it is possible to predict accidents with extremely good accuracy, which is important for reducing their number.

The same group of French authors, with exactly the same database, continued research and risk analysis at level crossings using Bayesian networks (BNs). In particular, causal structural constraints have been introduced in order to establish a BN risk model for the purpose of combining empirical knowledge and statistics, thus enabling the identification of effective causality and avoidance of inappropriate structural links. Furthermore, advanced and reverse conclusions based on risk determination using BN models were derived to predict the occurrence of accidents and incidents at level crossings and to quantify the degree of contribution of various factors and impacts, in order to determine risk factors. In addition, analysis of the model’s impact strength was performed to examine the impact strength of various causal factors on the occurrence of road accidents and incidents [42].

In the last few years, several scientific papers have appeared in which researchers have determined models for assessing the risk of accidents at level crossings. A known technique of mathematical programming for measuring the efficiency of complex entities with diverse inputs/outputs is the data Envelopment Analysis (DEA) software, DEA-Solver LV8.0, which allows one to determine whether the Decision-Making Unit (DMU) is effective or not. This technique was used in the analysis of the efficiency of 12 operational units representing the nearest cities where accidents occurred at road crossings in the Republic of Serbia in the period from 2005 to 2014, with the aim of reducing the number of accidents [43]. Another study [44] was conducted in the Republic of Serbia with the aim of identifying the necessary parameters that quantify the risks associated with railway crossings, where the available statistical models that are commonly used (Poisson, NB, ZIP and ZINB) were analyzed. These models were obtained using a gradual AIC. The obtained models were then compared using the Vuong test. In the mentioned research [44], a new measure for risk was introduced—empirical risk. In Bosnia and Herzegovina, for eight level crossings on the Šamac–Doboj line, crossings were evaluated using the Novel Integrated Fuzzy MCDM—fuzzy PIPRECIA (pivot pairwise relative criteria importance assessment) model [45], i.e., the assessment was performed using the fuzzy MARCOS method (measurement of alternatives and ranking according to compromise solution). The level of safety of level crossings was determined on the basis of determining the final weight values of the criteria, in order to achieve sustainable railway traffic management.

Research conducted in North Carolina (USA) was based on a model for predicting the risk of collisions at level crossings in relation to the class of rail and track [46], while another comprehensive study conducted in Florida (USA) recommended a new model for predicting danger, called the Florida priority index formula, for ranking/setting priorities at level crossings (highway/railroad). The Florida Priority Index formula provides a more accurate ranking of level crossings compared to alternative methods. The Florida priority index formula estimates the potential danger of a given level crossing based on the average daily traffic of road vehicles and trains [47]. The competitive risk model [48] is a special type of survival analysis designed to correctly assess the marginal probability of an incidence outcome when multiple causes of failure are possible. This method is intensively used in medical research to study the death of patients that can be attributed to competitive events such as cardiovascular and non-cardiovascular causes. Modeling the competitive risks of accidents at level crossings shows the ability to simultaneously identify risk factors and the marginal probability of the severity and occurrence of collisions. The model is designed to identify and summarize those factors that contribute to the probability of accident risk at level crossings in North Dakota. However, the study does not suggest the effectiveness of countermeasures.

## 3. Model of Heterogeneous Queuing System

### 3.1. Model Basics: Maximum Risk, Entropy and Chaos

The term “risk” is first found in Homer’s famous poems The Iliad and The Odyssey, 9th century AD. However, in these poems, the etymology of the word “risk” needs to be considered from the point of view of numerous synonyms of the word “radical” (Greek alphabet: ριζικό), which abounds in the poems of the Iliad and Odyssey, especially in terms of warrior uncompromisingness. Another more probable etymological source of risk is in the Greek word root (Greek alphabet: ριζκα), which was associated with the danger that threatened ships from underwater rocks. This associative source of the origin of “risk” is more probable if the terrestrial conditions of navigation of old are taken into account—sailing along the coast or along well-known cliffs, capes or islands, exclusively by day with good visibility. Terrestrial navigation was performed with the obligatory use of a depth gauge (a rope with knots weighed by a stone—the meaning of risk prevention). Under these navigational conditions, invisible underwater rocks (analogous to the invisible root below the earth’s surface) with unknown tidal intensities could have caused fatal damage to the ship. The primitive depth gauge thus received the title of the first means of risk assessment. Night sailing was made possible only with the advent of astronomical navigation in the 6th century BC, for which the famous Thales of Miletus is credited. Today, risk is considered to be the effect of uncertainty on goals (ISO31000 “Risk management—Principles and guidelines” standard, 2009).

Systemic chaos, in terms of complete disorder and confusion, can be caused under conditions of maximal entropy. Depending on the structure, some systems can achieve the goal even in conditions of maximal entropy, while other systems enter the progression of chaos and complete erosion of the system.

Traffic as a dynamic stochastic system has the desired value of zero risk—without accidents, i.e., without danger to human life and health and without degradation of the vehicles value, goods and roads. This ideal is inclined to the concept of safety in maximally regulated vehicle flows. Dominantly, traffic systems are homogeneous in an aggregate sense. A solid base is for road and rail traffic, liquid represents water traffic and gaseous represents air traffic system. Of these systems, the possible intersection of roads and flows at the same level can be realized on the land/ground—between the road and the rail traffic system. The level crossing serves traffic flows from two traffic systems and therefore must be considered as a heterogeneous system. Numerous concepts for accident prevention at level crossings are described in Section 2.

Contrary to the ideal of zero risk, the level crossing can be introduced into a state with maximal risk. This situation is achieved by declaring identical flow priorities in the absence of traffic signals as information carriers. It is clear that in real conditions, this mode of operation is prohibited. However, in virtual conditions, this mode can quantify the maximal risk.

In real conditions, after the realization of an accident at level crossings, the level crossing system was introduced into an unstable state. The work of the level crossing is obligatorily suspended, and the traffic flows of railway and road vehicles are brought to a halt. The consequences of the accident are fixed and procedures are carried out, which return the level crossing system to a stable, working condition. Otherwise, the level crossing system would reach a state of chaos. In theoretical conditions, virtual accidents do not stop traffic flows. Thus, theoretically, the level crossing system is introduced to chaos with the possibility of successive virtual accidents.

Objective risk can be obtained from the quotient of recorded real accidents n_real_ and the intensity of traffic flow of railway and road vehicles. There are a number of important parameters that are synthesized in this quotient. These are average values of the speed and length of trains and road vehicles, the non-stationary stochastic structure of road vehicle flow and the usual deterministic structure of railway vehicle flow, geometric characteristics of the level crossing (angle between the railway and road, horizontal and vertical curves), the dynamics of meteorological conditions, etc. The conditions of railway and road traffic and their impact are very difficult to statistically extract from the obtained value of objective risk. Let us denote the objective, real risk by *r_real_*.

Theoretically, the maximal risk of a level crossing is measurable. It is also obtained from the quotient of recorded virtual–theoretical accidents *n_theor_* and the intensity of traffic flow of railway and road vehicles. This leads to the risk interval, from complete reliability with zero risk to the maximal theoretical risk for given traffic flow intensities. If we denote the probability of the theoretical maximal–critical risk with *r_theor_*, objective, real risk is always in the range of theoretical limits 0 ≤ *r_real_* ≤ *r_theor_*.

Another important question remains: How to get the theoretical maximal risk? The answer to this question comes down to the induction of maximal entropy in the service system. Fortunately, the answer to this question is simple: By applying an exponential distribution, because of all the continuous distributions, it has the largest entropy. Therefore, it is enough to measure the intensity of traffic flows, the average length of trains and road vehicles and calculate the average occupancy of the level crossing by rail and road vehicles based on their average speeds. The parameters of the average occupancy of the level crossing can easily be included in the exponential distribution of the known function and density, for the parameter λ, which represents average daily intensity flows of a road or railway vehicle λ > 0, λ ∈ R(1) and time *t*, for the known parametric characteristics M(T) = λ^−1^, D(T) = λ^−2^ [49]:(1)$$F(t,\lambda )=\{\begin{array}{cc} 1-e^{-\lambda t} & t\geq 0\\ 0 & t<0 \end{array},\;f(t,\lambda )=\{\begin{array}{cc} \lambda e^{-\lambda t} & t\geq 0\\ 0 & t<0 \end{array}$$

Continuous entropy, or “differential entropy”, is a concept that expands the idea of “Shannon’s entropy”, as a measure of average predictability of the outcome of a random event with a continuous distribution of probabilities, i.e., measures of uncertainty established in the famous debate between Claude Shannon (2016–2001) and John von Neumann (1903–1957). In the example of the exponential distribution, the continuous entropy *h_e_*(*t*) is given by (2):(2)$$h_{e}(t)=-\int_{0}^{\infty }\lambda e^{-\lambda t}ln(\lambda e^{-\lambda t})dt=-ln(\lambda )+1$$

The application of the exponential distribution in the level crossing system without priority provides the maximal theoretical risk, precisely due to the known fact that of all probabilistic distributions, the exponential has the highest continuous entropy. In this sense, all probabilistic systems based on the exponential distribution are indeterminate to the greatest extent, which is the foundation of discrete processes based on Denis Poison’s distribution (1781–1840), and Cony Pelma’s theorem (1907–1951) proves the elementary importance of exponential distribution in Poison’s processes. This fact is contained in the “memoryless” property of exponential distribution, which has already found application in the field of tests for generating independent random numbers [50].

### 3.2. Ideal Level Crossing Queuing System

A level crossing can be considered as a queuing system that serves vehicles from two traffic systems: Railway and road. Therefore, the queuing system of a level crossing has a heterogeneous structure. In ideal safety conditions, without accidents, the level crossing queuing system is shown in Figure 1. This system has complementary occupancy states with unconditional priority for railway vehicles.

The probability of occupancy of the level crossing cumulants of time by railway vehicle is equal, “*p*”, with the geometric interpretation given in Figure 2. Probability “*p*” is analogous to the quotient of time cumulants *t_i_ i* ∈ [1, *n*] occupancy of the level crossing with priority railway traffic and the observed time T.

Complementarily, the access of road vehicles to the level crossing is realized with probability *q* = 1 − *p*, intensity *λ_c_*. The service of road vehicles is realized with intensity *μ_c_*.

The initial state of the Markov ergodic system is X_0,0_ without access to railway and road vehicles. The system has a defined priority of railway vehicles, which approach the road crossing with intensity *λ_t_*, and are served with intensity *μ_t_*.

When the level crossing is occupied by a railway vehicle, the system is in state X_0,1_. All road vehicles that approach the crossing with intensity *λ_c_*, and due to the defined priority of railway traffic, fill the system states in the order X_0+1,1_, X_0+2,1_, … X_0+k,1_. Ideally, the number of places in a row is unlimited (∞).

After the passage of the railway vehicle over the zone of the level crossing, the service with the intensity *μ_t_* has been performed. Regardless of the number of accumulated road vehicles, due to the *memoryless* property of exponential distribution, the system switches from state X_0+k,1_ to state X_k,0_, and the service of road vehicles with intensity *μ_c_* begins.

In case the road vehicle approaches the road crossing with intensity *λ_c_*, and the state is not conditioned by the priority of railway vehicles, the system changes the condition from X_0,0_ to X_1,0_, while serving with intensity *μ_c_* without the possibility of forming a queue.

The presented system is idealized from the side of safety. It prevents the simultaneous access of vehicles of the two systems to the road crossing, i.e., there is no condition X_1,1_, which is a collision of railway and road vehicles.

### 3.3. Queuing System for the Calculation of Maximal Risk

The issue of calculating the value of the maximal risk can be solved by applying a heterogeneous queuing system with the following parameters:*λ_c_* average daily intensity flows of road vehicle.*λ_t_*, average daily intensity flows of railway vehicle.*μ_c_* average daily service intensity of road vehicles.*μ_t_*, average daily service intensity of railway vehicles.

The key geometric parameters of the model for the service intensity calculation (Figure 3) are:*l_c_* average length of road vehicles.*L_c_* length of the critical distance for road vehicles, which is equal to the average frontal width of the train.*l_t_* length of the critical distance, which is equal to the average train length.*L_t_* length of the critical distance for railway vehicles, which is equal to the width of the level crossing.

The service intensity of road vehicles is calculated from quotient (3) where *v_c_* is the average speed of road vehicles over the critical distance *l_c_ + L_c_*:(3)$$\mu_{c}=\frac{l_{c}+L_{c}}{v_{c}}$$

The service intensity of trains is calculated from quotient (4) where *v_t_* is the average speed of railway vehicles over the critical distance *l_t_ + L_t_*:(4)$$\mu_{t}=\frac{l_{t}+L_{t}}{v_{t}}$$

In order to achieve maximum entropy, i.e., for the calculation of the maximal risk, an exponential distribution was adopted for the obtained parameters. The declared probabilities of the states are as follows (Figure 4):X_0,0_ road crossing is without vehicles.X_1,0_ level crossing is at a critical distance and serves only road vehicles.X_0,1_ level crossing is at a critical distance and serves only railway vehicles.X_1,1_ road crossing serves both road and rail vehicles at the same time. This condition has a dual accident status: A railway vehicle may run into a road vehicle, or a road vehicle may run into a railway vehicle.

A graph of the heterogenous level crossing system’s states of service for railway and road vehicles for the calculation of maximal risk is presented in Figure 5.

All intensities are obligatorily exponential in order to achieve maximal entropy and to calculate the maximal risk. The flows of road and railway vehicles are Poisson’s. Due to the well-known property of summing the intensity of Poisson flows, which was proved by Raikov’s theorem [51], the distribution of two-way flows of road vehicles can be distributed without restriction on the width of the road *L_t_*, which also represents the length of critical distance for railway vehicles. The system is Markovian, and the probabilities are calculated directly in the stationary mode [52]. The probabilities of the states for the Markovian stationary system are calculated from the initial Equation (5):(5)$$\begin{array}{l} p_{0,0}^{\prime }\left(t\right)=0=-\lambda_{c}p_{0,0}-\lambda_{t}p_{0,0}+\mu_{c}p_{1,0}+\mu_{t}p_{0,1}\\ p_{1,0}^{\prime }\left(t\right)=0=-\mu_{c}p_{1,0}-\lambda_{t}p_{1,0}+\lambda_{c}p_{0,0}+\mu_{t}p_{1,1}\\ p_{0,1}^{\prime }\left(t\right)=0=-\mu_{t}p_{0,1}-\lambda_{c}p_{0,1}+\mu_{c}p_{1,1}+\lambda_{t}p_{0,0}\\ p_{1,1}^{\prime }\left(t\right)=0=-\mu_{t}p_{1,1}-\mu_{c}p_{1,1}+\lambda_{t}p_{1,0}+\lambda_{c}p_{0,1} \end{array}$$

The norming condition of the system is (6):(6)$$p_{0,0}+p_{1,0}+p_{0,1}+p_{1,1}=1$$

By grouping the values of intensities (5) with unknown probabilities of states, respectively, *p*_0,0_, *p*_1,0_, *p*_0,1_ and *p*_1,1_ give equation system (7):(7)$$\begin{array}{l} \left(-\lambda_{c}-\lambda_{t}\right)p_{0,0}+\mu_{c}p_{1,0}+\mu_{t}p_{0,1}=0\\ \left(-\mu_{c}-\lambda_{t}\right)p_{1,0}+\lambda_{c}p_{0,0}+\mu_{t}p_{1,1}=0\\ \left(-\mu_{t}-\lambda_{c}\right)p_{0,1}+\mu_{c}p_{1,1}+\lambda_{t}p_{0,0}=0\\ \left(-\mu_{t}-\mu_{c}\right)p_{1,1}+\lambda_{t}p_{1,0}+\lambda_{c}p_{0,1}=0 \end{array}$$

By elimination with the obligatory use of norming condition (6), the probabilities of the system’s states are (8):(8)$$\begin{array}{ccc} & p_{1,0}=\frac{\lambda_{c}\mu_{t}}{\left(\mu_{c}+\lambda_{c}\right)\left(\mu_{t}+\lambda_{t}\right)} & \\ p_{0,0}=\frac{\mu_{c}\mu_{t}}{\left(\mu_{c}+\lambda_{c}\right)\left(\mu_{t}+\lambda_{t}\right)} & & p_{1,1}=\frac{\lambda_{c}\lambda_{t}}{\left(\mu_{c}+\lambda_{c}\right)\left(\mu_{t}+\lambda_{t}\right)}\\ & p_{0,1}=\frac{\mu_{c}\lambda_{t}}{\left(\mu_{c}+\lambda_{c}\right)\left(\mu_{t}+\lambda_{t}\right)} & \end{array}$$

The probability of the state of maximal risk is equal to the probability of simultaneous occupancy of the level crossing with both railway and road vehicles *p*_1,1_. Since all the parameters for calculating the state probability X_1,1_ i.e., intensities in the numerator, are positive and greater than zero *λ_c_* > 0 ∧ *λ_t_* > 0, and the product in the denominator is always greater than zero (*μ_c_* + *λ_c_*)(*μ_t_* + *λ_t_*) > 0, the maximal risk always converges!

The number of realized accidents, for the purposes of comparison, can be reduced to units of road vehicles that are statistically recorded at the real level crossing *n_real_* in time interval T, which is the basis for calculating the accident intensity per vehicle, or the probability that any road vehicle will have an accident *p_real_*, which is equal to (9):(9)$$p_{real}=\frac{n_{real}}{T\cdot 365\cdot \lambda_{c}}$$
where *λ_c_* is the previously explained unit intensity of road vehicles’ access in a day, while T is the time period expressed in years for which the statistical records are analyzed. The real probability of an accident has a dual possibility of expression through the intensity of railway vehicles’ access as well. However, due to the usually higher intensity of road vehicles at level crossings *λ_c_* >> λ_t_, in further research, we shall adopt the intensity of road traffic as a reference for the calculation. Equation (9) gives the probability that an arbitrary road vehicle will, in real conditions, participate in an accident.

In theoretical conditions, the heterogeneous system in state X_00_ with probability p_00_ serves 0 road vehicles and in state X_01_ (when the level crossing is occupied only by railway vehicle) with probability p_01_ also serves 0 road vehicles. This means that the entire flow of road vehicles is serviced in states X_1,0_ and X_1,1_ with probabilities *p*_1,0_ and *p*_1,1_, respectively. Therefore, it is necessary to divide the flow of road vehicles into the number of vehicles that were serviced at the level crossing, but did not participate in an accident, and the number of vehicles found in critical condition X_1,1_, which is the number of vehicles that participated in theoretical accidents. Based on the ergodicity of the system, the distribution is proportional (10). The obtained value has the dimension of accidents in one day (due to the fact that the flow intensity is declared per day).
(10)$$\lambda_{c}=\underset{\begin{array}{l} nonaccident\\ traffic\;flow \end{array}}{\underbrace{\frac{p_{1,0}}{p_{1,0}+p_{1,1}}\lambda_{c}}}+\underset{n_{theor}}{\underbrace{\frac{p_{1,1}}{p_{1,0}+p_{1,1}}\lambda_{c}}}$$

The conditions when a level crossing system is introduced into chaos with the possibility of successive virtual accidents and where *n_theor_* in the time interval T (number of years) are the foundation for calculating the intensity of theoretical, virtual accidents per vehicle, or the probability that arbitrary road vehicle will be involved in a virtual accident. According to the analogous pattern for *p_real_* (9), the theoretical maximal probability of an accident per vehicle *p_theor_* is obtained (11):(11)$$p_{theor}=\frac{n_{theor}}{T\cdot 365\cdot \lambda_{c}}\left[\frac{accident}{day\frac{vehicle}{day}}\right]=\frac{\frac{p_{1,1}}{p_{1,0}+p_{1,1}}T\cdot 365\cdot \lambda_{c}}{T\cdot 365\cdot \lambda_{c}}=\frac{p_{1,1}}{\left(p_{1,0}+p_{1,1}\right)}$$

The maximal theoretical risk is an important reference value from which the reliability of the operation of the level crossing is obtained, which takes into account all the previously mentioned parameters, and above all, the flow intensities of road and railway vehicles. In statistically recorded accidents, i.e., indirectly through *p_real_,* all other characteristics of the work of the level crossing are implicitly contained. The synthetic reliability of the road crossing is given by (12):(12)$$R=\frac{p_{theor}-p_{real}}{p_{theor}}$$

The complementary value of reliability is the risk of the level crossing (13)
(13)r=1−R

## 4. Model Testing on Selected Level Crossings

For the specific application of the model, two level crossings on the lines of “Serbian Railways Infrastructure” were selected. The frequencies of road and railway traffic, technical and geometric data, were obtained by measurements and counting on the spot. Part of the statistical data on the number of accidents at level crossings was used from the OC for SP Ruma, where one author was involved in the organization and supervision of traffic on the observed part of the railway. Part of the data was simultaneously used and analyzed from a project conducted through the study “Research of traffic safety in the areas of level crossings” by the Road Traffic Safety Agency of the Republic of Serbia during 2018, in cooperation with professors of the Department of Traffic from the Faculty of Technical Sciences in Novi Sad and a large number of experts in this field [53].

The first level crossing is “Buđanovci”, which is located on the local road L-1 (Ruma-Budjanovci-Nikinci-Platičevo), 3 + 285 km along the regional railway Ruma-Šabac-Rasputnica Donja Borina-state border-(Zvornik Novi). The level crossing is passive and is provided with traffic signs both on the road and the railway along with the zone of required visibility (visibility triangle). The satellite view with geographical coordinates of the road crossing is given in Figure 6.

All stationary and dynamic parameters of the level crossing are shown in Table 1. The average length of road vehicles was obtained based on the structure of the flow (cars, trucks, buses, agricultural vehicles with machinery). The number of recorded accidents in the period between 2007 and 2017 at this road crossing is 6.

The second level crossing is “Platičevo”, which is located on the state road Novi Sad-Irig-Ruma-Šabac and 21 + 465 km along the regional railway Ruma-Šabac-Rasputnica Donja Borina-state border-(Zvornik Novi). The level crossing is active and is equipped with half-barriers as well as light and sound signals. This level crossing is managed from the station, exclusively by the train dispatcher when securing the route. The satellite view with geographical coordinates of the level crossing is given in Figure 7.

All stationary and dynamic parameters of the level crossing are shown in Table 2. The average length of road vehicles was obtained based on the structure of the flow (cars, trucks, buses, agricultural vehicles with machinery). The number of recorded accidents in the period between 2007 and 2017 at this road crossing is 2.

For an easier review, data on ratio synthetics reliability (*R*) and quotients *p_real_* and *p_theor_*, as well as risks (*r*), for both observed level crossings are given in Table 3 and Figure 8, as well as Table 4 and Figure 9, respectively.

Based on the obtained synthetic parameters that take into account all the static and dynamic attributes of the observed level crossings, we come to an important objective answer that has been in the subjective zone so far: How much riskier is the “Buđanovci” level crossing than “Platičevo” level crossing? The ratio of their risks is (14): (14)$$\Delta =\frac{r_{Budjanovci}}{r_{Platicevo}}=\frac{0.00038286959}{0.00001337065}=28.6351$$

### Discussion

An ideal safety state, as we have shown in Figure 1 and Figure 2 and in part of the work in point 3.2, the risk of an accident is *r* = 0 because there is no danger that there will be a collision of road and railway vehicles. The system represented as the ideal system has complementary occupancy states with unconditional priority for railway vehicles.

In reality, this ideal situation is disturbed by the risk of potentially dangerous accidental occurrences. In this paper, a theoretical model is set up, where, with the mathematical apparatus presented above, we come to the calculation number of virtual, theoretical accidents, the calculation of the risk of accidents and complementary reliability for the observed level crossings.

As the first comparative result, the obtained value of 28,6351 times higher safety at the “Platičevo level crossing” than the safety at the “Buđanovci” level crossing is somewhat expected. Here we should keep in mind the different levels of security and different stationary and dynamic parameters and the much higher flow of road vehicles at the “Platičevo” level crossing.

Based on expression (10) at the level crossing in Budjanovci, the daily number of road vehicles that do not participate in an accident is 8,030,969, and the number of vehicles that participate in an accident is 3,903,146, while in Platičevo, based on expression (10), the flow of road vehicles that do not participate in an accident is 7,590,744, and the number of vehicles that participate in an accident is 3,725,564.

The “Platičevo” level crossing is an active level crossing and it has a higher level of security in relation to the level crossing “Buđanovci” and a smaller number of real accidents realized in the observed period. Regardless of the far higher flow of road vehicles at the “Platičevo” level crossing, according to the probability of a real accident *p_real_*, every 15,313,210th (1/*p_real_*) vehicle will participate in an accident, while at the “Buđanovci” crossing, every 540,017.5th (1/*p_real_*) vehicle will participate in an accident, which contributes to the obtained test results of the model.

The “Platičevo” level crossing is an active level crossing, where the users of the crossing are protected or are warned of an approaching train by activation of the device, in the case where it is not safe for the user to cross the crossing. In this case, control is manual, and the level crossing device is activated by an authorized person of the railway infrastructure, when they receive information through the means of communication that ensures safety, as part of the safe traffic organization. At the “Platičevo” level crossing, there is a lowering barrier when the train approaches the level crossing in order to prevent the crossing of road vehicles or pedestrians, which significantly affects the level of safety and security of this crossing. The model is presented in a valid way through the calculation of synthetic reliability and comparison of the maximal risk for both crossings, which explicitly stated which crossing is safer and how many times.

As a result of this work, a proposal can be made to raise the level of security, in terms of raising the level of security at the “Buđanovci” level crossing. Until the moment of the essential technical-technological solution, it is possible, in order to preserve and raise the level of traffic safety at the “Buđanovci” level crossing, to take some less demanding and cheaper measures such as:The installation of vibrating lanes in order to “calm down” road traffic when approaching the area of the road-rail crossing.Improving the visibility at the level crossing (all visibility triangles must be absolutely provided).Paving the surface of the road–rail crossing (level crossing) with different strong colors, lighting the crossing with reflectors and installing additional light signals and lanterns that will warn road users that they are about to encounter the level crossing, etc.

The above measures are proposed as a temporary solution until such time as the Public Management for Road Infrastructure and Public Management for Railway Infrastructure reach a decision on the provision of financial resources for automated level crossings (which is risky) on the basis of the above analysis, statistical data on the number of accidents, lightly and seriously injured persons at the specified level crossing for the period from 1996 to 2020 (Table 5) as well as the on-site inspection of the situation at the level crossing (Figure 10 and Figure 11).

## 5. Conclusions

The proposed model and the preliminary results do not carry the pretense imperative of the final form of the model. The application of maximal probabilistic entropy has been premiered and successfully performed in order to quantify the maximal risk that has been proven convergent. With the ideal value of the minimal risk, which is always the ideal of absolute safety, the necessary engineering interval is declared: From zero risk to maximum risk. It is omnipotent and can be applied to homogeneous road traffic systems at intersections, or in other traffic systems, and to conflicted flows in general.

The comparison of risks at level crossings is based on a statistic parameter: The number of accidents. The installation of new, or the reconstruction of existing, level crossings significantly reduce this value. For known geometric characteristics and flow intensities, the proposed model provides the possibility of a preliminary calculation of the maximal risk as the starting reference value of subsequent safety elaborations. In addition, the model can easily compare level crossings of different safety levels or level crossings of the same level of safety installed by different manufacturers.

The advantage of this model for infrastructure managers and road managers lies in the fact that by using the methodology of performing reliability analyses and risk comparisons at all observed level crossings, it is possible to solve the problem of determining priorities when choosing level crossings where it is necessary to raise safety. This represents a great contribution, keeping in mind the limited financial resources and the unlimited desire to raise the level of traffic safety.

The proposed model explicitly included only the basic parameters. Implicitly, there are a large number of probabilistic parameters at road crossings: Average sensory and motor abilities of drivers, driving culture and habits, reliability of signaling and safety devices, road quality at level crossings, visibility (meteorological), time of day, air temperature and humidity, devices acoustic audibility, the severity of accidents (number of injured or killed), etc. All these parameters are implicitly covered through only one statistic parameter: The number of accidents. In future work and research, these parameters can be introduced into the model by special analytical functions, with one obligatory principle: Regardless of their deterministic or probabilistic structure, the maximal probabilistic entropy must be preserved by applying an exponential distribution.

## Figures and Tables

**Figure 1 entropy-23-01230-f001:**
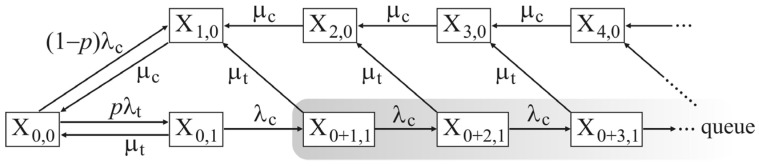
Level crossing queuing system with unconditional priority.

**Figure 2 entropy-23-01230-f002:**
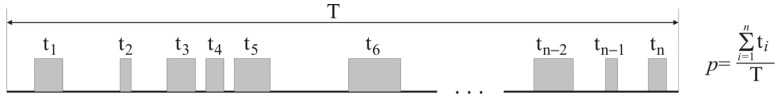
Geometric interpretation probabilities of level crossing occupancy.

**Figure 3 entropy-23-01230-f003:**
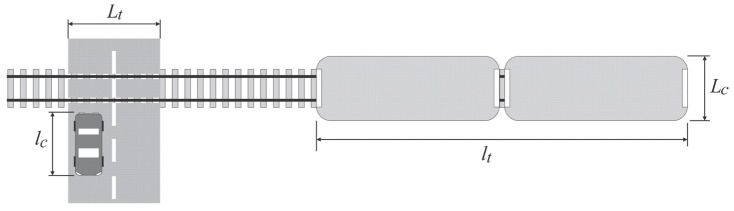
The key geometric parameters of the model.

**Figure 4 entropy-23-01230-f004:**
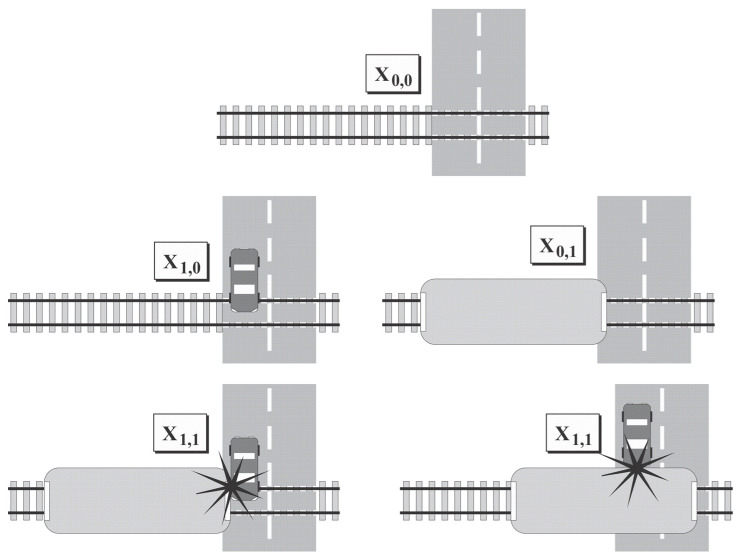
Possible states of heterogenous level crossing system.

**Figure 5 entropy-23-01230-f005:**
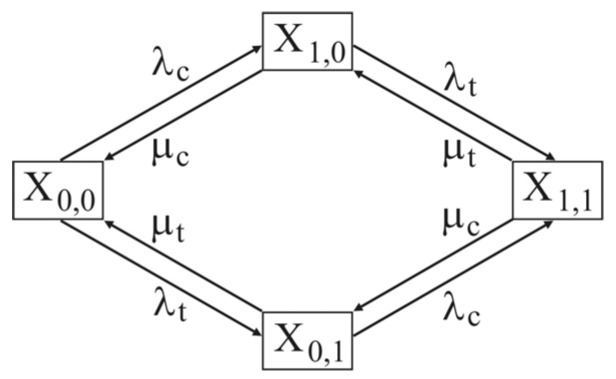
The graph of states.

**Figure 6 entropy-23-01230-f006:**
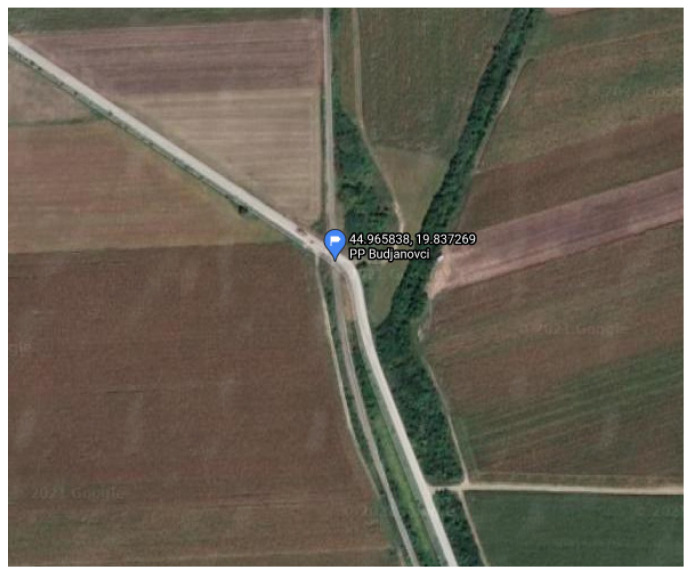
Satellite image of the “Buđanovci” level crossing.

**Figure 7 entropy-23-01230-f007:**
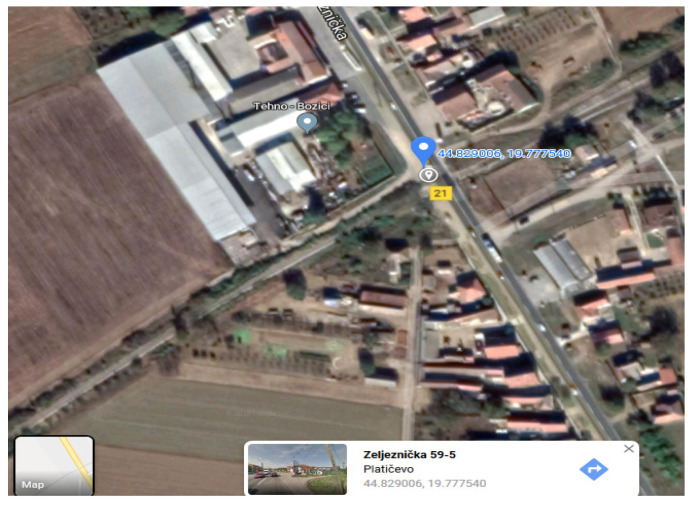
Satellite image of the “Platičevo” level crossing.

**Figure 8 entropy-23-01230-f008:**
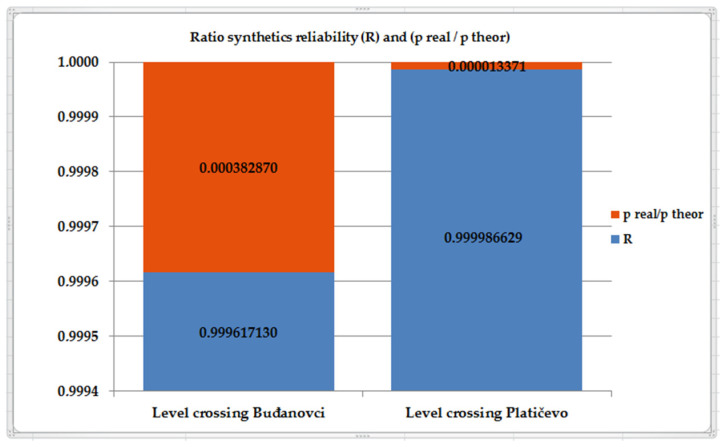
The graphical representation ratio of synthetic level crossing reliability (*R*) and *p_real_*/*p_theor_*.

**Figure 9 entropy-23-01230-f009:**
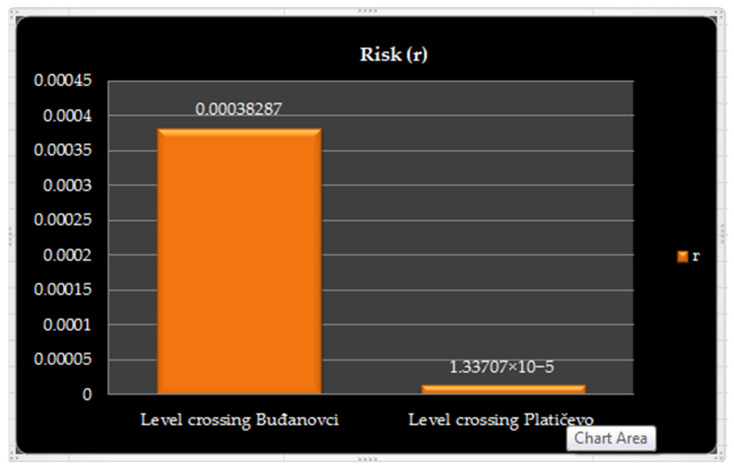
The graphical representation calculated values of risk (*r*).

**Figure 10 entropy-23-01230-f010:**
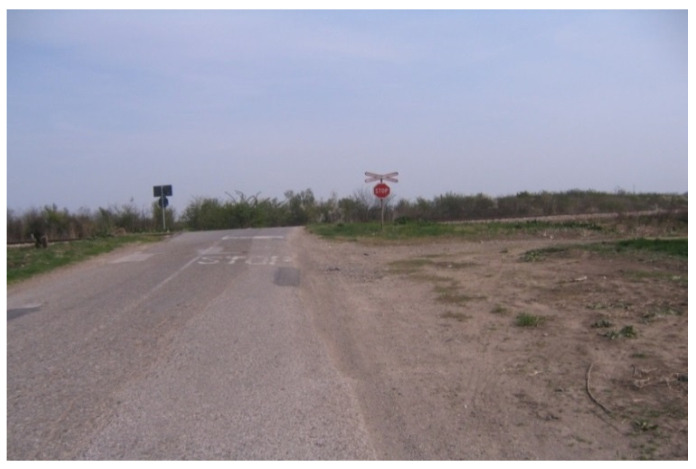
Passive “Budjanovci” level crossing faded horizontal signalization.

**Figure 11 entropy-23-01230-f011:**
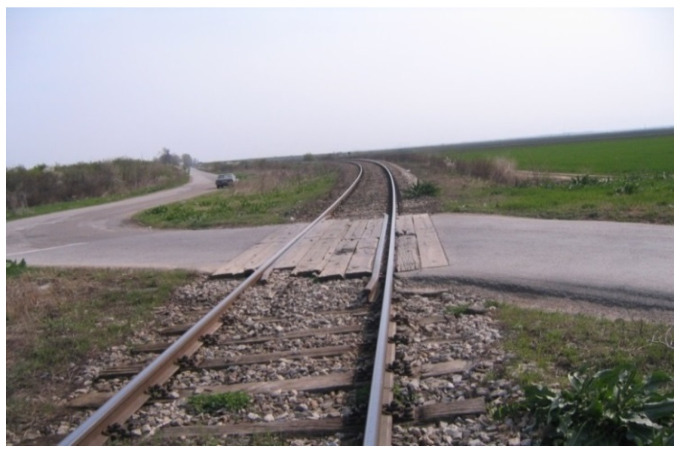
Road geometry for the passive “Budjanovci” level crossing, with horizontal and vertical curves and poor visibility.

**Table 1 entropy-23-01230-t001:** Stationary and dynamic parameters of the “Buđanovci” level crossing.

Stationary and Dynamic Parameters	Value
Average train speed over the level crossing (km/h)	70
Average speed of road vehicles over the level crossing (km/h)	32
*λ_t_* (trains/day)	23
*λ_c_* (road vehicles/day)	807
*l_t_* (average train length in m)	350
*L_t_* (length of railway vehicle over critical distance in m)	5.00
*l_c_* (average length of road vehicle in m)	6.80
*L_c_* (length of road vehicle over critical distance in m)	2.80
Train service intensity *μ_t_* = (*l_t_* + *L_t_*)/*v_t_*	4,732.394
Road service intensity *μ_c_* = (*l_c_* + *L_c_*)/*v_c_*	80,000.000
The probability of a real accident *p_real_*	0.000001851792
The probability of a theoretical accident *p_theor_*	0.004836612535
Synthetic level crossing reliability (*R*)	0.99961713041
Risk (*r*)	0.00038286959

**Table 2 entropy-23-01230-t002:** Stationary and dynamic parameters of the “Platičevo” level crossing.

Stationary and Dynamic Parameters	Value
Average train speed over the level crossing (km/h)	34
Average speed of road vehicles over the level crossing (km/h)	70
*λ_t_* (trains/day)	23
*λ_c_* (road vehicles/day)	7628
*l_t_* (average train length in m)	350
*L_t_* (length of railway vehicle over critical distance in m)	8.50
*l_c_* (average length of road vehicle in m)	5.20
*L_c_* (length of road vehicle over critical distance in m)	2.80
Train service intensity *μ_t_* = (*l_t_* + *L_t_*)/*v_t_*	4786.192
Road service intensity *μ_c_* = (*l_c_* + *L_c_*)/*v_c_*	102000.000
The probability of a real accident *p_real_*	0.000000065303
The probability of a theoretical accident *p_theor_*	0.004884064551
Synthetic level crossing reliability (*R*)	0.99998662935
Risk (*r*)	0.00001337065

**Table 3 entropy-23-01230-t003:** Calculated values of (*R*) and *p_real_*/*p_theor_*.

	Level Crossing Buđanovci	Level Crossing Platičevo
*R*	0.999617130	0.999986629
*p_real_*/*p_theor_*	0.000382870	0.000013371

**Table 4 entropy-23-01230-t004:** Calculated values of risk.

Risk	Level Crossing Buđanovci	Level Crossing Platičevo
*r*	0.00038287	1.33707 × 10^−5^

**Table 5 entropy-23-01230-t005:** Review of the number of accidents, injuries and deaths on the “Budjanovci” level crossing from 1996 to 2020.

Number of Accidents, Injuries and Deaths	Value
Number of accidents in the period from 23 years (1996–2020)	12
Number of persons who were deaths at the level crossing (1996–2020)	2
Number of persons who were seriously injured at the level crossing (1996–2020)	4
Number of persons who were lightly injured at the level crossing (1996–2020)	1

## Data Availability

Data used in this paper can be found from https://www.putevi-srbije.rs/index.php/брoјање-саoбраћаја (accessed on 12 March 2021).

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
