# Peer review of "Model for Risk Calculation and Reliability Comparison of Level Crossings"

_entropy, 2021, doi:10.3390/e23091230_

Round 1
Reviewer 1 Report
The authors did significant work on their paper. The paper presents a model for risk calculation and reliability comparison of level crossings. The basic idea is that the exact reliability of the level crossing is calculated from the real and maximal risk ratio, which enables their further comparison to raise the level of safety. First, the authors presented a literature review regarding models for predicting accidents and incidents at level crossings. Then, the authors proposed a new model for risk calculation and reliability comparison of level crossings which were then tested on two representative ones. My only suggestion would be that authors could add discussion as a separate chapter before the conclusion. I will leave this to the authors to decide; it is not an obligation.
Author Response
We thank Reviewer 1 for the commendable review.
The proposal for the development of the discussion will be considered after the correspondence with the reviewer 2.
Please understand. Thanks
Authors
Reviewer 2 Report
Please see the attached word file.

Author Response
The large difference between the views of the reviewers points to the next "round" of consideration.
Reviewer 2 categorically, in the first paragraph, states that "this paper was written with a lack of logic". Please, we are not "beginners"
First, the etymology of risk is still unclear. That is why we have relied on a historical introduction. For practical application, we have given the definition of ISO31000 "Risk management - Principles and guidelines" standard, 2009
There is an obvious misunderstanding between reviewer 2 and the authors. It is eloquently visible in paragraph 6 of the reviewer: "In line 304, ... is it reasonable to assume there is no possibility of forming a queue? Is your model able to handle this situation: when not all road vehicles have crossed the level cross , a railway vehicle comes? What will happen in your model?"
No, there is no queue in the model! All vehicles in conditions of chaos, without any traffic rules, approach the road crossing. Virtual accidents are realized in a mathematical model. This is the state (1.1) of queueing system!
The intensity of accidents depends on the intensity of road and rail vehicle flows.
The flow structure is exponential necessarily. The reason is the maximum entropy of the exponential distribution. The application of any other continuous probabilistic distribution (Normal, Gamma, Erlang, etc.) gives less systemic entropy, less chaos.
Maximum entropy implies maximum chaos, ie. the maximum number of accidents resulting in the maximum risk at the road crossing. The maximum risk (probability of the condition (1,1,)) is multiplied by the road flow and the number of vehicles affected by the accident is obtained. This number is compared with the real number of accidents that is statistically recorded.
The key focus of the manuscript is the non-existence of queues for road vehicles at the road crossing - due to the existence of chaos, due to the calculation of maximum risk.
I ask reviewer 2 to unravel our idea once again. The idea is completely new, unconventional. We have an understanding for his report, but we expect that with additional explanations he will find ways to accept it and understand our idea based on maximum entropy, please.
The previous "rough" review is not adequate to the manuscript.
Reviewer 3 Report
Dear Authors.
Congratulations, nice and correct article, a great job.
With respect,
Reviewer
Author Response
Dear Madam/Sir
Thanks for the explicit comment
We would like to inform you that we have expanded the discussion in accordance with the suggestions of reviewer 1.
Authors
Round 2
Reviewer 2 Report
Basically, the authors should respond to how they consider my comments point-by-point. However, they only replied to the comments that they want to reply to. For example, comments 1, 2, 4, 5, 8 were not considered. Even comments 6 and 7 were not well addressed.
Additionally, we can find a clear definition of risk in the context of infrastructure security in the literature (e.g.,Flage, R., T. Aven, and Enrico Zio. "Alternative Representations of Uncertainty in System Reliability and Risk Analysis: Review and Discussion." ESREL 2008. 2008; Flage, R., T. Aven, and Enrico Zio. "Alternative Representations of Uncertainty in System Reliability and Risk Analysis: Review and Discussion." ESREL 2008. 2008;Aven, Terje. Risk analysis. John Wiley & Sons, 2015.). Therefore, the authors claimed that "the etymology of risk is still unclear" is not convincing. The first comment in my report should be clearly addressed. Moreover, if the definition and physical meaning of the maximal risk is still unclear, why can you use it?
In short, the issues in the first version have not been well addressed yet. This work should be rejected without any further evaluation due to its poor contribution and bad quality.